# Vitamin D and Its Role on the Fatigue Mitigation: A Narrative Review

**DOI:** 10.3390/nu16020221

**Published:** 2024-01-10

**Authors:** Ippolita Valentina Di Molfetta, Laura Bordoni, Rosita Gabbianelli, Gianni Sagratini, Laura Alessandroni

**Affiliations:** 1Chemistry Interdisciplinary Project, School of Pharmacy, University of Camerino, Via Madonna delle Carceri, 62032 Camerino, Italy; ippolitav.dimolfetta@studenti.unicam.it (I.V.D.M.); laura.alessandroni@unicam.it (L.A.); 2Unit of Molecular Biology and Nutrigenomics, School of Pharmacy, University of Camerino, Via Gentile III da Varano, 62032 Camerino, Italy; laura.bordoni@unicam.it (L.B.); rosita.gabbianelli@unicam.it (R.G.)

**Keywords:** vitamin D, oxidative stress, neurotransmitters, chronic fatigue, musculoskeletal disorders

## Abstract

Vitamin D has historically been associated with bone metabolism. However, over the years, a growing body of evidence has emerged indicating its involvement in various physiological processes that may influence the onset of numerous pathologies (cardiovascular and neurodegenerative diseases, rheumatological diseases, fertility, cancer, diabetes, or a condition of fatigue). This narrative review investigates the current knowledge of the pathophysiological mechanisms underlying fatigue and the ways in which vitamin D is implicated in these processes. Scientific studies in the databases of PubMed, Scopus, and Web of Science were reviewed with a focus on factors that play a role in the genesis of fatigue, where the influence of vitamin D has been clearly demonstrated. The pathogenic factors of fatigue influenced by vitamin D are related to biochemical factors connected to oxidative stress and inflammatory cytokines. A role in the control of the neurotransmitters dopamine and serotonin has also been demonstrated: an imbalance in the relationship between these two neurotransmitters is linked to the genesis of fatigue. Furthermore, vitamin D is implicated in the control of voltage-gated calcium and chloride channels. Although it has been demonstrated that hypovitaminosis D is associated with numerous pathological conditions, current data on the outcomes of correcting hypovitaminosis D are conflicting. This suggests that, despite the significant involvement of vitamin D in regulating mechanisms governing fatigue, other factors could also play a role.

## 1. The Fatigue

Fatigue is a symptom related to a great variety of physiological and paraphysiological conditions, and in many of these, it represents an epiphenomenon. Its manifestations are both peripheral and central, presenting as a condition of weakness and torpor associated with a reduction in muscle strength. This peripheral condition may be due to a pathological condition that affects either the muscle fibers or its control by motor neurons [1]. On the contrary, mental fatigue can be associated with a variable degree of cognitive deterioration. In both cases, the result is a negative impact on quality of life. Quantitatively assessing fatigue is challenging due to numerous subjective variables and different classifications. According to work by Engberg et al. (2017), women have higher fatigue scores than men, and among men, higher socioeconomic status and greater physical activity are associated with lower levels of fatigue [2]. According to Behrenas et al. (2023), fatigue may be distinguished by the state of fatigue because the latter derives from two factors closely connected to each other: the fatigue objectively attributable to doing something and the fatigue that the patient perceives subjectively [3]. Chronic fatigue is an epiphenomenon of both central forms, such as neurological diseases relating to the central, peripheral, and autonomic nervous systems, and peripherical ones, such as those affecting the neuromuscular junction and metabolic diseases, as well as rheumatologic diseases; furthermore, this symptom also prevails among autoimmune diseases (rheumatological diseases, type 1 diabetes, chronic fatigue syndrome, or celiac disease) [4,5]. Although fatigue is a very disabling symptom that strongly compromises the quality of life, and although its pathophysiology is the subject of numerous studies, the mechanisms that lead to the onset of this symptom are currently not perfectly clear. Many involved features have been identified, including cytokines, some neurotransmitters, factors associated with oxidative stress, and many factors involved in their regulation; among them, there is vitamin D.

Despite substantial evidence for the potential role of vitamin D in regulating fatigue across various pathological conditions, current clinical practices have overlooked the potential impact of vitamin D supplementation on such conditions. We thus conducted this review with two aims in mind: (1) to summarize the knowledge about the effect of vitamin D levels on the occurrence and/or progression of chronic fatigue-related diseases, as determined by in vitro, in vivo models, and human clinical trials; and (2) to provide a rationale for further investigations (such as randomized clinical trials) to translate current evidence into clinical practice. For this review, we identified scientific studies published up to 2023 in the databases of PubMed, Scopus, and Web of Science, using search terms that combine “vitamin D”, “vitamin D3”, or “hypovitaminosis D” with the various pathological conditions.

## 2. Vitamin D

Vitamin D is a group of lipid-soluble precursors that need to be converted into the active form; the most important compounds in this group are cholecalciferol (D3) and ergocalciferol (D2). D2 differs from D3 because it has a double bond between C22 and C23 and a methyl group at C24 of the side chain. The D3 form is specific to animal species, while the D2 can be found in some vegetable matrices [6]. The main sources of vitamin D are two: an endogen and an exogen. The first and most important one consists of the endogenous synthesis at the level of human skin via a UVB light photochemical reaction [7]. The second source, which is less decisive, consists of food intake. Food sources include mushrooms and reindeer lichen for D2 and fish, in particular fish liver oils, for D3 [8]. Since most mammals are able to synthesize vitamin D3 if adequately exposed to sunlight, the consequence is that it is not considered a vitamin in the strict sense but more as a hormone. Several in-depth revisions of vitamin D sources (both from food and as supplements) can be found in the literature [8].

In a work by Sintzel et al. (2018), the great difference between the self-produced vitamin D and the dietary intake was investigated [9]. They calculated that during summer, a 20 min whole body UVB ray exposure session of a Caucasian person may ensure more than 10,000 IU of produced vitamin D3. In contrast, diet provides a very small and less significant contribution, which consists only of 40–400 IU for each average meal.

Both vitamin D2 and D3 need to be activated via two consequential hydroxylation reactions: firstly, at the liver level, where cholecalciferol is converted to calcifediol (25-hydroxycholecalciferol, 25(OH)D) because of a 25-hydroxylase (CYP2R1), and then at the renal level, where calcifediol is further hydroxylated to form calcitriol (1,25-dihydroxycholecalciferol, 1,25(OH)_2_D) by 1α-hydroxylase (CYP27B1). Calcitriol is the biologically active form of vitamin D3 in humans [10]. Furthermore, extra-renal paracrine activation has been demonstrated in >10 extrarenal organs [11].

The main mechanisms that prevent hypervitaminosis resulting from UVB over-exposure are represented both by 24-hydroxylation performed by the enzyme 24-hydroxylase (CYP24A1) and by the degradation of 25(OH)D and 1,25(OH)_2_D produced by the UVB rays themselves [12].

The three main steps, 25-hydroxylation, 1α-hydroxylation, and 24-hydroxylation, are performed in the endoplasmic reticulum by Cytochrome P450 mixed function oxidases located in CYP2R1, which seems to be the major 25-hydroxylase, or in the mitochondria by CYP27A1, CYP27B1, and CYP24A1 [10]. Indeed, the active form of vitamin D is generated not only in renal tubular cells but also in extrarenal target tissue cells, and its functions are autocrine, paracrine, and endocrine.

The vitamin D metabolites are transported in blood bound primarily to vitamin D binding protein (DBP) (85–88%) and albumin (12–15%) and explicate their function in the target tissues interacting with the Vitamin D Receptor (VDR).

The chemical difference between D2 and D3 results in the different affinity of D2 for DBP (D-binding protein), weaker in D2 than in D3 [13]. Both active forms of vitamin D2 and vitamin D3 have comparable affinity for VDR [10].

Vitamin D exerts its biological effects by binding to its intranuclear receptor (VDR); after this binding, a heterodimeric complex with the retinoid X receptor is obtained.

This complex binds VDREs (vitamin D response elements) that are specific sequences distributed in target genes near promoters and involves many coregulatory complexes; all these factors determine the transcriptional output, leading to the expression of numerous specific gene products [14].

The multiplicity and diversity of the effects of vitamin D depend on the different localization of the VDRs, which are present in multiple target organs [15]. VDRs have been identified in both the cytoplasm and nucleus, associated with different nuclear membranes. For this reason, the effects of vitamin D can be both genomic and non-genomic [16].

The pathophysiology underlying the relationship between fatigue and the regulatory phenomena exerted by vitamin D is quite complex and concerns both the phenomena linked to the reduction in oxidative stress and inflammation and neurobiological effects associated with both growth factors and neurotransmitters (Figure 1).

### 2.1. Vitamin D’s Impact on the Cytokines: Oxidative Stress and Inflammation

As previously mentioned, oxidative stress and inflammation are major pathways regulating fatigue onset [17].

Many mechanisms are involved in these complex processes, such as redox reactions, ROS (reactive oxygen species) production, and normal mitochondrial functions. Vitamin D was studied as an important regulatory factor that may control these pathways [18].

The activation of vitamin D is enhanced in conditions of cellular stress [19], and the supplementation of vitamin D is able to improve mitochondrial oxidative function in skeletal muscle and reduce oxidative stress in the mitochondria [20]. In fact, during oxidative stress conditions, there is an increase in the release of many factors, such as nuclear factor kappa β (NFkβ), inducible NO synthase (iNOS), and cyclooxygenase-2 (COX-2), leading to the enhancement of nitrosative stress and consequently the damage of fatty acids and proteins at the membrane level, all underpinning chronic fatigue [21].

An important vitamin D-regulated mitochondrial pathway consists of the control of Nrf2/PGC-1α-SIRT3 [22]; in this process, peroxisome proliferator-activated receptor—coactivator 1α (PGC-1α) interacts with Nrf2, generating a complex that modulates SIRT3. Vitamin D ligand–receptor binding increases the transcription of Nrf2, a major regulator of redox signaling, which exerts antioxidant activities by enhancing the expression of numerous genes involved in these pathways [23,24,25,26].

Another mechanism of protection from oxidative damage concerns the enhanced expression of the anti-aging protein Klotho [23]. Klotho is a gene that encodes for an anti-aging protein with various functions, including increasing the resistance to oxidative stress [27].

Moreover, vitamin D-VDRE binding determines a favorable influence on mitochondrial respiration and inhibits the overproduction of ROS [28]. In fact, its chronic deficiency is connected to the formation of ROS and, hence, oxidative damage [18].

Another important role of vitamin D consists of the regulation of the expression of gamma-glutamyl transpeptidase (g-GT), which contributes to the synthesis of a reduced form of glutathione (GSH), essential for the activity of glutathione peroxidase (GPX). It is one of the cell’s enzymatic antioxidant systems and can increase the activity of other enzymes such as glucose-6-phosphate dehydrogenase (G6PD), glutamate–cysteine ligase (CGL), and glutathione reductase, leading to a further increase in GSH [29].

Remarkably, vitamin D can also act at the epigenetic level. Vitamin D can influence the epigenome in multiple ways, such as increasing genomic VDR binding, influencing CCCTC binding factor (CTCF) binding and the formation of topologically associated domains (TADs), affecting the binding of pioneer transcription factor and changing histone modifications and chromatin accessibility [30].

Also, because of its epigenetic effects, vitamin D is a major regulator of the immune system and of inflammatory processes [31]. A causal link between vitamin D and inflammation has been demonstrated concerning both the synthesis and release of anti-inflammatory cytokines and direct actions on the cells of the immune system. According to Carlberg et al. (2019), vitamin D modulates the epigenome of immune cells both in vitro and in vivo, in particular monocytes and their differentiated subtypes [30].

Reducing the differentiation of Th1 lymphocytes, the active form of D, can reduce the secretion of inflammatory cytokines (IL-2, IFNγ, and TNF-α). At the same time, it can promote Th2-type differentiation, increasing the secretion of anti-inflammatory cytokines (IL-4, IL-5, and IL-10) [32].

Furthermore, the active form of vitamin D promotes monocyte proliferation and the IL-1 and cathelicidin expression by macrophages and inhibits the proliferation of B cells and their differentiation into plasma cells, consequently reducing immunoglobulins production [3].

Many immune system cells (dendritic cells, monocytes/macrophages, and T and B cells) express 1α-hydroxylase, so many immune responses can be attributable to VDR [33]. In fact, activated T cells can upregulate the VDR, and vitamin D is increased, not only by the expression of VDR but also by cytochrome P450 and CYP27B1 in macrophages and monocytes [3].

In addition, pathogenic T cells produce IL-17, and this is supported by IL-23. The stimulation of this IL-23 cytokine is strongly inhibited by the D-VDR interaction [34]. Interestingly, some proinflammatory cytokines (INL-1, INL-6, and INF γ) are often increased in patients with chronic fatigue [35].

### 2.2. Vitamin D’s Interaction with Neurotransmitters

It was demonstrated that vitamin D can affect both growth factors and neurotransmitter activity. Concerning the growth factors, for example, a relationship has been demonstrated between vitamin D and NGF (Nerve Grow factor) [36,37].

Vitamin D3 was reported to increase the levels of NGF, GDNF (glial cell-derived neurotrophic factor), and the expression of p75NTR, a tumor necrosis factor receptor expressed during early neuronal development and in adults, in specific pathological conditions [38]. Despite that, no effects on other neurotrophins emerged [39].

Central fatigue relates to the activity of several neurotransmitters, including serotonin, dopamine, acetylcholine, angiotensin II, norepinephrine, and nitric oxide; among them, vitamin D seems to influence dopamine and serotonin [40]. Vitamin D can carry out its action at the brain level, as VDR has been found in many brain areas [41].

Central fatigue results from an imbalance of dopamine and serotonin within the central nervous system [35,42,43]. In this scenario, vitamin D has proven to be a key factor in the regulation of neurogenesis and differentiation processes; in fact, according to recent research, a chronic administration of vitamin D can lead to an increase in the ability to release dopamine [44]. The authors assessed that vitamin D is a powerful regulator acting on the distribution of synapses, vesicle proteins, and a number of dopaminergic synapses. Moreover, it can improve the function of tyrosine hydroxylase in cells.

Many authors suggest a role for vitamin D as an adjuvant in the treatment of specific pathologies; for instance, vitamin D was studied as an important factor in maintaining the trophism of dopaminergic neurons in Parkinson’s disease [45]. The same relationship in other morbid conditions, such as attention-deficit/hyperactivity disorder [46], or in neuropsychiatric diseases was assessed [47]. The relationship between vitamin D and fatigue does not only concern its influence on dopamine pathways but also serotonin, as its association with the onset of fatigue was demonstrated. In fact, an alteration of serotonin resulted in an alteration of fatigue [48]. This connection was also underlined by Zawadzka et al. (2021), who emphasized how fatigue is linked to an altered central relationship between serotonin and dopamine [49].

The vitamin D–serotonin biochemical relationship derives from vitamin D’s influence on the tryptophan hydroxylase transcription, as the hydroxylation of tryptophan is a limiting step in serotonin biosynthesis [50,51]. From a molecular point of view, the ligand–receptor binding would trigger the activation of many genes, including TPH2, which is responsible for vitamin D synthesis and tryptophan metabolism [52].

Moreover, the relationship between the two neurotransmitters, dopamine and serotonin, is of importance as fatigue is related to increased serotonergic activity and decreased dopaminergic activity [40].

Despite this evidence supporting the key role of vitamin D in the modulation of neuronal responses, further data on the correlation between plasma vitamin D and Parkinson’s disease or other neuronal disorders should be acquired to support its use within preventive strategies of neurodegeneration.

### 2.3. Vitamin D’s Impact on Other Molecular Pathways

Among the other molecular pathways involved in vitamin D activity, it was demonstrated that it can induce the activity of voltage-gated chloride and calcium channels [19]. Vitamin D plays a neuroprotective role by modulating L-type voltage-dependent calcium channels via a genomic and a non-genomic mechanism, both via the modulation of substances such as calbindin and by upregulating neurotrophic factors (NGF, GDNF, and NT-3); the modulation of calcium channels performs clear functions on neuronal excitability [53].

With regard to the voltage-dependent chloride channels, according to our current knowledge, the modulation by vitamin D has been demonstrated only at the Sertoli cell level and at the osteoblastic level [54,55].

Another mechanism of action of vitamin D in reducing fatigue consists of its influence on some characteristics of red blood cells, including the ability to deform when they pass through capillaries; the deformability of red blood cells, which appears to be altered in patients with chronic fatigue, was correlated with plasma levels of 25(OH)D and other substances such as alanine aminotransferase (ALT), aspartate aminotransferase (AST), and C-reactive protein (CRP) [56].

## 3. Hypovitaminosis D: Causes, Symptoms, and Impact on Fatigue

Vitamin D deficiency occurs when plasma levels of 25(OH)D are below 20 ng/mL [57]; levels between 21 and 29 ng/mL define insufficiency, and levels over 30 ng/mL are considered sufficient [58].

A daily dose of 600–800 IU of vitamin D was reported to be enough to guarantee good bone health; however, to have 25(OH)D stable plasma levels higher than 30 ng/mL, an increased intake is necessary (1000–2000 IU) [3].

Many factors influence plasma vitamin D levels, including the time of day, time of year, latitude (in higher latitudes, vitamin D is not produced year-round), skin factors (the presence of clothing and the type and color of the fabric), the use of sunscreen and sun protection factor, the amount of melanin (i.e., African Americans, and dark-skinned populations in general, have lower levels of vitamin D), the presence of damaged skin (scar tissues have a lower capacity to synthesize active vitamin D), the skin temperature (an increase in skin temperature corresponds to a greater speed of skin vitamin synthesis), the number of vitamins ingested and absorbed (many pathologies reduce absorption), and possible obesity (which is associated with lower serum vitamin D concentrations) [59].

Chronic hypovitaminosis D is associated with cardiovascular diseases and metabolic dysfunctions and could be considered an important comorbidity or a risk factor for premature death [60,61]. In fact, inverse relationships have been reported with vitamin D adequacy, with reduced all-cause mortality and cancer [62]. Other consequences of hypovitaminosis D can be headaches and muscle pains [63].

Moreover, there is evidence supporting the role of vitamin D (plasma levels above 30 ng/mL) in the protection against arterial hypertension and a possible association with glucose metabolism [15].

Since vitamin D levels have been linked to many serious pathological conditions, both neurological and autoimmune, the opportunity for vitamin D supplementation in these patients has been considered. However, data supporting this hypothesis are controversial. The role of vitamin D in fatigue related to specific diseases and conditions is explored in further sections (Figure 2).

### 3.1. The Point of View of Fibromyalgia

Fibromyalgia is a systemic and chronic painful condition in which a relevant and prevalent symptom is fatigue. Although it is an increasingly morbid condition, and although it is a condition that often comes to the attention of rheumatologists (after arthritis), its pathogenesis is not completely clear. Despite that, two important factors are involved in it concerning the alteration of neurotransmitter balance and of neuronal inflammatory processes leading to a condition of peripheral hyperalgesia [64].

A recent meta-analysis demonstrated an association between serum vitamin D levels and fibromyalgia, reporting lower blood levels of vitamin D in patients with fibromyalgia compared to control groups [65].

In general, the study of a causal link with various factors, including vitamin D, is very challenging, so nowadays, there is no consistent evidence regarding the linkage between vitamin D and fibromyalgia.

Despite that, the correction of hypovitaminosis D with an improvement in fatigue in fibromyalgia was clearly correlated by Bilal et al. (2009), showing very encouraging results on the amelioration of several fibromyalgia ACR (American College of Rheumatology) criteria and on the “chronic fatigue” symptom. According to these authors, the improvement reached 97.6% [66].

The correction of hypovitaminosis, if present, was reported to be useful both to reduce the extent of osteoporosis and to improve strength and muscular performance [67].

In scientific literature, most of the evidence about hypovitaminosis correction in controlling fibromyalgia focuses on pain control, which represents a seriously disabling symptomatology in these patients. Fatigue is also considered a critical symptom. A work by Solmaz et al. (2015) showed interesting results regarding the correlation between serum vitamin D levels and the extent of pain and fatigue [68]: their assessments were based both on the visual analog scale (VAS) and the functional assessment of BASFI and HAQ. Furthermore, laboratory analysis and Widespread Pain Index (WPI) were evaluated. A recent systematic review of randomized control trials by Lombardo et al. (2022) showed that vitamin D supplementation reduced pain in 6 of 434 studies considered from 1990 to 2022, highlighting a link between the deficiency of vitamin D and muscle pain in patients with fibromyalgia [69]. Thus, despite promising findings from Bilal et al. [66], current evidence on the hypothesis that vitamin D supplementations may specifically control fatigue in fibromyalgia is too scarce to support its usage in clinical practice.

### 3.2. The Point of View of Multiple Sclerosis

Multiple sclerosis (MS) is a neurological demyelinating disease that falls under the autoimmune diseases. Among the various epidemiological criteria of this disease, low vitamin D values have often been identified [70].

Normal plasma quantities of vitamin D may reduce the risk and influence the course of the disease; in fact increasing 25(OH)D levels by 50% may reduce the chance of contracting MS by about 50%; at the same time, the increase in plasma vitamin D values is correlated with a decrease in the rate of new active lesions visible via Magnetic Resonance Imaging (MRI) [3].

Knippenberg et al. (2014) assessed a reduction in symptoms such as fatigue linked more to sunlight exposure than to plasma vitamin D increase [71]. These authors opted for the evaluation of depressive and anxious symptoms based on the HADS (Hospital Anxiety and Depression Scale), while fatigue was evaluated using the FSS (Fatigue Severity Scale), and cognitive performance via the PASAT (Paced Auditory Serial Addition test) and neurological disability by the EDSS (Expanded Disability Status Scale).

In contrast, Beckmann et al. (2020) concluded a notable improvement in fatigue and quality of life with the increase in plasma vitamin D values [72]. Also, in this work, the assessments were based on EDSS (Expanded Disability Status Scale), while in the case of fatigue and quality of life, respectively, the Fatigue Severity Scale (FSS) and the MS-related quality of life inventory were used (MSQOLI).

This discrepancy between the various data in the literature was underlined in a recent meta-analysis where it was assessed that, since there is no agreement between the literature data regarding the use of vitamin D use in fatigue symptom control, it is advisable to use vitamin D only upon prescription [71]. Moreover, no agreement about vitamin D supplementation doses was reported. In fact, some studies kept the supplementary recommendations only in case of deficiency; others used doses lower than 600 IU/day to avoid toxicity from hypervitaminosis; and others associated high doses with a good outcome, and others with a bad one [71,72,73]. In general, evidence of supplementation effectiveness was reported mainly in patients with low plasma vitamin D levels. Moreover, further studies are needed to define the impact of vitamin D on MS-related fatigue and to optimize suitable doses and treatment times.

### 3.3. The Point of View of Rheumatological Diseases

The chapter on rheumatological diseases is represented by many autoimmune disorders, with a common denominator represented by fatigue. Also, in this case, conflicting data were found in the scientific literature.

The main limitation in this field is represented by the fact that the published studies are more concentrated on the correlation between vitamin D and the pathogenesis of diseases rather than its relationship to fatigue in these patients. A review concerning the treatment of fatigue with vitamin D supplementation in SLE also reaches similar conclusions [32]. An administration of at least 800 IU/day in patients with low levels of vitamin D was recommended by Ruiz-Irastorza et al. (2010), with eventual dose adjustment made on the basis of starting plasma vitamin D values [74].

Roy et al. (2014) recommend a plasma vitamin D test in patients with symptoms of fatigue as low serum vitamin D level is generally recognized in these patients, and a significant reduction in fatigue severity was reported after its correction [75]. In contrast, Jelsness-Jørgensen et al. (2020) stated that in rheumatoid arthritis, there is no correlation between fatigue and plasma vitamin D levels [76].

An improvement in inflammatory markers was reported after vitamin D supplementation in patients with rheumatoid arthritis, while in the case of systemic lupus erythematosus, fatigue may be reduced with the same supplementation [75].

### 3.4. The Point of View of Myasthenia Gravis

Myasthenia gravis is a relatively uncommon autoimmune disorder where fatigue is one of the predominant symptoms. To our knowledge, there are few studies relating vitamin D to the control of fatigue in myasthenia gravis.

A study by Askmark et al. (2012) correlated plasma vitamin D values with myasthenia gravis, and demonstrated that in these patients, compared to controls, plasma vitamin D value was significantly lower [77]. Another case study on a 49-year-old woman revealed that despite the conventional therapy for myasthenia gravis, she experienced a symptom improvement only when she was administered high doses of vitamin D (80,000 IU) [78]. In contrast, recent research reported that not only is there no causal relationship between myasthenia gravis and low vitamin D levels but the latter does not even represent a risk factor [79]. Thus, the hypothesis of a correlation between myasthenia gravis and vitamin D is currently supported by limited and contrasting evidence in small cohorts of individuals. Further studies in bigger cohorts are necessary for a better understanding of the role of vitamin D in this pathological condition.

### 3.5. The Point of View of Elderly Age

The correlation between fatigue in old patients and vitamin D is well defined. Compared to all the other conditions described, there is greater agreement regarding the opportunity to correct hypovitaminosis D in elderly people.

A strong association of mental and physical fatigue, evaluated by FSS (Fatigue Severity Scale) with low D plasma level was assessed, with gender differences was identified regarding both physical and mental fatigue with a greater incidence in males, so a supplementation of D vitamin is recommended in these patients [58]. An implementation of plasma vitamin D and calcium levels can be correlated with muscle fatigue via the regulation of the biosynthesis of creatine kinase, lactate dehydrogenase, troponin I, and hydroxyproline via a mechanism that controls the production of free radicals [80]. Moreover, considering fatigue as an element of frailty, a relationship of inverse proportionality between plasma vitamin D values and the extent of frailty was demonstrated [81]. However, Roy et al. (2014) underlined how, despite the evidence on the beneficial effect of vitamin D supplementation, there is no clarity on the doses and methods of administration of this vitamin [75].

Thus, a significant body of evidence supports the beneficial role of vitamin D in contrasting fatigue in the elderly, even though further studies addressing time and dose administrations are warranted.

### 3.6. The Point of View of Cancer

Low plasma levels of vitamin D have often been identified in cancer patients with advanced disease and cachexia or fatigue [82]. Correction of hypovitaminosis D significantly improves the severity of fatigue symptoms [83].

A study by Khan et al. (2010) on women with breast cancer treated with letrozole focused on the association between fatigue, osteoarticular pain, and vitamin D values; the cut-off was set is 66 ng/mL 12 weeks from the start of treatment with 50,000 IU of vitamin D per week. Women with levels higher than the cut-off demonstrated improvement and elimination of symptoms, unlike others with vitamin D below this cut-off [64]. Furthermore, not only was an improvement in fatigue following the administration of vitamin D observed but also a good safety profile was identified regarding the association of vitamin D and chemotherapy. Despite promising evidence, there is a paucity of studies that specifically investigate fatigue control as a primary outcome following vitamin D supplementation in cancer patients. Furthermore, it remains unclear whether the potential beneficial effects on fatigue are universally exerted by vitamin D independently of cancer type and disease status or if there are specific conditions warranting the recommendation of such supplementation. Consequently, additional studies addressing these critical issues are warranted to provide a more nuanced understanding of the relationship between vitamin D supplementation, fatigue, and cancer.

### 3.7. The Point of View of Chronic Fatigue Syndrome

Chronic fatigue syndrome, better known as myalgic encephalopathy, is defined as a condition with an incompletely clear etiology characterized by debilitating tiredness lasting more than six months, associated with at least four other symptoms: non-restorative sleep, impaired memory and/or concentration, episodes of headaches, post-exertional malaise, muscle pain and polyarthralgia, and sore throat [84]. The most accredited pathogenetic hypothesis is linked to phenomena triggered by viral diseases and conditions such as hypocortisolism, but in reality, this condition must be classified as a complex pathology with repercussions in the multifactorial cognitive behavioral field [85]. At the moment, to our knowledge, no drug therapy has proven effective, so the therapy is based on cognitive behavioral therapy and graded physical therapy. A work by Earl et al. (2017) investigated the relationship between this syndrome and plasmatic vitamin D values without identifying hypovitaminosis condition in these patients [86]. This evidence appears in contrast with what was reported in previous studies such as that of Berkovitz et al., which, on the contrary, demonstrated low plasma levels of vitamin D in these patients [87]; in fact, one of the conclusions of this study concerned the hypothesis of recommending vitamin D supplementation in these patients and sun exposure, as well as the use of foods rich in vitamin D. A more recent, large study investigated the relationship between chronic fatigue or low energy levels and low circulating levels of 25OHD [88]; the data emerging from this study suggest that a clinically relevant protective effect of vitamin D on fatigue is unlikely, so these authors believe that the correction of hypovitaminosis D is unlikely to be protective against fatigue.

### 3.8. The Point of View of Parkinson’s Disease

In Parkinson’s disease (PD), fatigue is a symptom that can occur early in the disease and also as a pre-motor symptom. In this context, the pathophysiological causes that determine fatigue are not known, and it does not respond adequately to pharmacological and surgical treatments [89]. In the literature, there are some works that have demonstrated an inverse relationship between plasma vitamin D values and Parkinson’s disease; among these, a work from Wang et al. (2015) proves an inverse correlation between PD and vitamin D levels [90]. The same conclusion of Ding et al. (2013) [91]. According to Pignolo et al. (2022), using the Unified Parkinson’s Disease Rating Scale (UPDRS) and the Hoehn and Yahr (H&Y) scale was observed: vitamin D deficiency appears to be associated with disease severity and progression; in fact, since the neuroprotective role is lacking, this could be linked to death of dopaminergic neurons [92]. Furthermore, it has been highlighted that VDR and 1α-hydroxylase are highly expressed in the substantia nigra [93]. Moreover, Pignolo a et al. (2022) suggest that, although there are no clear data in the literature, as the potential benefits are greater than the potential limited risks, it is possible that this supplementation will be evaluated in the near future [92]. Despite this recommendation, it must be admitted that evidence connecting vitamin D and fatigue control in PD is currently very limited.

### 3.9. The Point of View of Neuropsychiatric Diseases

Many metabolites of vitamin D have been identified in cerebrospinal fluid, and they have been shown to cross the blood–brain barrier.

At the nervous system level, the region with the greatest density of vitamin D receptors is represented by the substantia nigra, which we know is also the region in which dopamine is most represented.

Other regions characterized by a high density of vitamin D receptors are represented by the external state of the granule cells of the prefrontal cortex and the hypothalamus. These reactors are also found in the cerebellum, in the thalamus, and in the hippocampus [94].

Vitamin D is implicated in gene regulation associated with neuroplasticity and neuroprotection phenomena. Considering this assumption, it may be legitimate to evaluate the link with possible vitamin D deficiency. Krisanova et al. (2019), in a preclinical study, explored the correlation between low vitamin D values and the functioning of the neurotransmitters GABA and Glutamate in the reagents, concluding that in case of hypovitaminosis D, there is a malfunction in the transport of these neurotransmitters [95].

According to Eyles et al. (2013), it was hypothesized that vitamin D deficiency in early life influences neuronal differentiation, axonal connectivity, dopamine ontogeny, and brain structure and function [49]. The correlation between low vitamin D values and fatigue in some psychiatric diseases is emerging.

It is also known that fatigue is a symptom frequently observed in patients suffering from neuropsychiatric diseases, but in this case, the pathophysiology is unclear [96]. Rylander and Verhulst (2013) demonstrated vitamin D deficiency (defined as levels < 30 ng/mL) in 75% of their psychiatric patient; in fact, vitamin D deficiency has been associated with some psychiatric conditions, autism spectrum disorder, and schizophrenia [97]. A study highlighted the correlation with major depression demonstrated low levels of vitamin D, but despite this, no impact on response to treatment compared to patients with normal vitamin D values [98]. Although there is evidence that identifies low levels of vitamin D in psychiatric patients, to the best of our knowledge, no certain data known to us concerns the correlation of improvement in fatigue with the correction of hypovitaminosis.

In the literature, the correlation between vitamin D and many neuropsychiatric diseases is explored, mainly in relation to the manifestation of the symptomatic process in these diseases [94,99].

### 3.10. The Point of View of Musculoskeletal Disorders

Due to vitamin D’s central role in the musculoskeletal system and, consequently, the strong negative impact on bone health in cases of vitamin D deficiency, a role of this vitamin in counterbalancing fatigue associated with musculoskeletal disorders can be hypothesized. Rickets, osteomalacia, osteopenia, primary and secondary osteoporosis, as well as sarcopenia and musculoskeletal pain, are all positively correlated with a vitamin D deficiency [100]. However, there is a great controversy regarding the appropriate vitamin D supplementation as both positive and negative effects on bone mineral density, musculoskeletal pain, and incidence of falls are reported. Results of observational studies assessing the relationship between vitamin D and musculoskeletal pain are somewhat inconsistent, with some, although not all, reporting a significant association between low vitamin D and chronic widespread pain in specific groups [101,102,103]. For example, supplementation with vitamin D (60,000 IU of vitamin D orally once weekly for 1 month, and then 60,000 IU once a month for the next 2 months) and calcium (1000 mg/day) decreased chronic non-specific musculoskeletal pain in a study conducted on 50 hypovitaminosis D patients [104]. A gender-specific association has been observed by McBeth et al. in a subset of the European Male Ageing Study cohort, showing a correlation between chronic widespread pain and low vitamin D in men at baseline [102].

However, even though persistent fatigue is a frequent complaint of individuals with chronic pain, pain, and fatigue are not always overlapping [105]; thus, specific evidence on the effect of vitamin D supplementation on fatigue associated with musculoskeletal disorders is necessary to demonstrate a potential application of this vitamin in the control of this symptom.

## 4. Conclusions

The present work aims at reviewing the state of the art on vitamin D effects on the mitigation of fatigue state. Important findings about the impact of vitamin D in specific fatigue-related molecular pathways, such as inflammation and oxidative stress, emerge from the literature.

Despite the promising potential effects of vitamin D on these pathways can be hypothesized, limited data on in vivo studies (especially in human cohorts) are currently present in the literature. In addition, where studies in human cohorts are available, conflicting findings emerge. Also, despite the fact that vitamin D deficiency is recurrent in patients affected by several different diseases, scarce data on the impact of supplementation on fatigue as an outcome have been collected until now. These limitations of the current knowledge on the potential impact of vitamin D on fatigue significantly constrain the translatability of research findings into clinical practice. Concerning the level of evidence in different pathophysiological conditions, results can be summarized as follows. A clear connection between fatigue and vitamin D in elderly people emerged. Also, a good response against fatigue in multiple sclerosis patients when supplemented with vitamin D has been documented. In contrast, very little and mostly conflicting evidence has been published so far regarding the other pathologies considered in this review (i.e., fibromyalgia, rheumatological diseases, myasthenia gravis, and cancer). Thus, according to the current level of evidence, it is not possible to support the usage of vitamin D supplementation on fatigue in these patients.

Since vitamin D supplementation might represent an easily accessible remedy to a serious condition, such as fatigue, which is also common to many severe pathologies, further studies on this topic should be conducted. In particular, given the limitations and biases of current studies as delineated throughout the manuscript, studies able to demonstrate a causal impact of vitamin D supplementation on fatigue mitigation (such as randomized controlled clinical trials) should be conducted in order to translate the evidence about vitamin D ability to control fatigue in clinical practice.

In conclusion, vitamin D supplementation appears as a promising tool that might help to mitigate the fatigue state, even if further scientific knowledge is needed to have a clear view on this topic.

## Figures and Tables

**Figure 1 nutrients-16-00221-f001:**
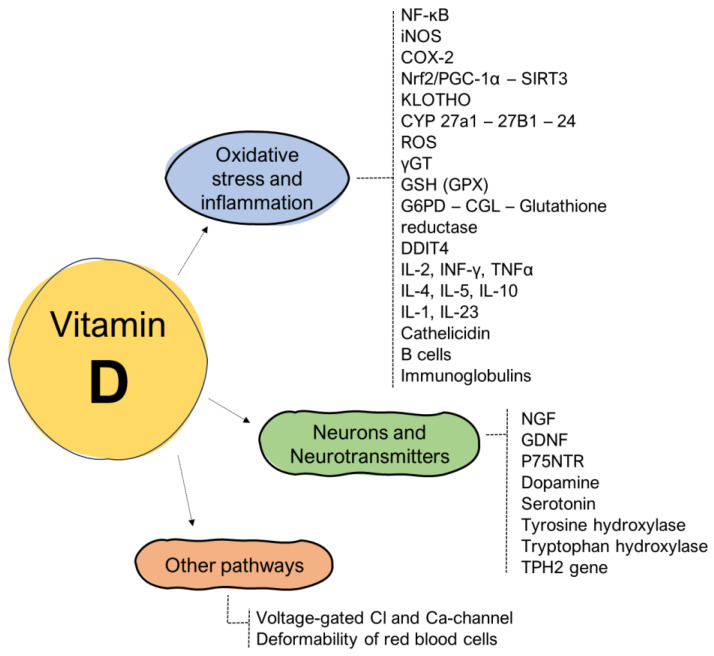
Influence of vitamin D on the pathogenetic mechanisms related to the onset of fatigue.

**Figure 2 nutrients-16-00221-f002:**
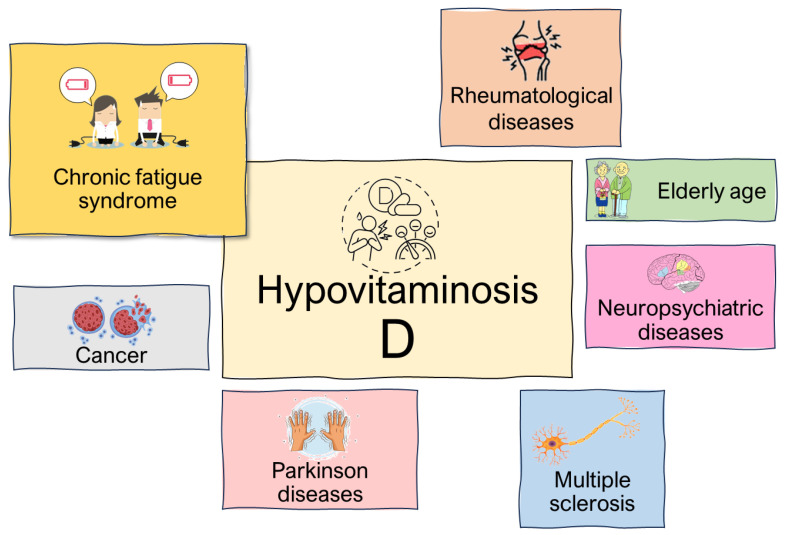
Physiopathological conditions affected by hypovitaminosis D.

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
