# Peer review of "Vitamin D and Its Role on the Fatigue Mitigation: A Narrative Review"

_nutrients, 2024, doi:10.3390/nu16020221_

Round 1
Reviewer 1 Report
Comments and Suggestions for Authors
This is a review paper about Vitamin D and its role in fatigue mitigation.
The authors reviewed the effect of vitamin D on fatigue relief based on numerous papers. As a review article, it was easy to read and well-structured.
This paper has potential information about the association between vitamin D and fatigue.
I propose a few revision comments, as shown below;
- The author mentions the association between neurotransmitters and vitamin D. It would be nice to see a mention of vitamin D and fatigue about Parkinson's or neuropsychiatric disease in section 3.
- In section 3, the authors reviewed the association between fatigue and vitamin D in each disease. Please show how fatigue was assessed in previous studies and the results for fatigue relief for each disease.
- In section 3, whether vitamin D improved fatigue because it improved disease symptoms or whether fatigue was assessed separately is unclear. Please provide a more specific description of these findings.
- The author mentioned that vitamin D affects fatigue mitigation through multiple mechanisms, and it would be nice to have an overall schematic picture of this.
Author Response
This is a review paper about Vitamin D and its role in fatigue mitigation.
The authors reviewed the effect of vitamin D on fatigue relief based on numerous papers. As a review article, it was easy to read and well-structured.
This paper has potential information about the association between vitamin D and fatigue.
We thank the reviewer for the positive comments on the paper and for the suggestions that strongly contributed to improve the overall quality of the manuscript. We addressed the questions as detailed below.
I propose a few revision comments, as shown below.
The author mentions the association between neurotransmitters and vitamin D. It would be nice to see a mention of vitamin D and fatigue about Parkinson's or neuropsychiatric disease in section 3.
According to reviewer suggestion two sections were added in the manuscript: one on Parkinson’s disease and vitamin D (section 3.8) and another on neuropsychiatric disease (section 3.9)
In section 3, the authors reviewed the association between fatigue and vitamin D in each disease. Please show how fatigue was assessed in previous studies and the results for fatigue relief for each disease.
We detailed the method used for fatigue assessment in section 3.1 (lines 297-303).
In section 3, whether vitamin D improved fatigue because it improved disease symptoms or whether fatigue was assessed separately is unclear. Please provide a more specific description of these findings.
We thank the reviewer for the suggestions, according to that, to make these findings more clear the section 3.7 was added.
The author mentioned that vitamin D affects fatigue mitigation through multiple mechanisms, and it would be nice to have an overall schematic picture of this.
According to this we added a schematic figure about the main pathways influenced by vitamin D (Figure 1).
Reviewer 2 Report
Comments and Suggestions for Authors
After reviewing the paper titled "Vitamin D and its Role on the Fatigue Mitigation," several key areas for improvement have been identified. Below is a brief introduction followed by a list of critiques:
Introduction:
The paper delves into the multifaceted role of Vitamin D, primarily focusing on its impact in mitigating fatigue. While the authors have comprehensively reviewed various pathophysiological mechanisms and the involvement of Vitamin D, certain critical aspects seem to be overlooked or insufficiently explored.
1. Lack of Research Methodology: The paper does not clearly outline the methodology used for the literature review. A detailed description of the databases searched, keywords used, inclusion and exclusion criteria, and the process of selecting articles would enhance the reliability and reproducibility of the review.
2. Omission of Key Literature on Vitamin D and Musculoskeletal Pain: The paper fails to address significant studies linking Vitamin D deficiency with musculoskeletal pain, which is closely related to fatigue. Inclusion of such studies would provide a more holistic understanding of Vitamin D's role in mitigating fatigue-related symptoms.
3. Insufficient Analysis of Gender and Age Differences: While the paper touches upon different population groups, it lacks a detailed analysis of how Vitamin D’s role in fatigue mitigation varies across different genders and age groups. Considering the physiological differences in Vitamin D metabolism, such an analysis would be pertinent.
4. Limited Discussion on Vitamin D Sources and Supplementation Guidelines: The paper does not sufficiently explore the different sources of Vitamin D (e.g., sunlight exposure, diet, supplements) and lacks specific guidelines or recommendations for Vitamin D supplementation, which would be valuable for both clinicians and patients.
Inadequate Exploration of the Role of Vitamin D in Chronic Illnesses: While some chronic conditions are discussed, the paper does not thoroughly explore the role of Vitamin D in the context of chronic illnesses known for causing fatigue, such as chronic fatigue syndrome and fibromyalgia.
5. Neglect of Recent Advancements in Vitamin D Research: The paper does not incorporate the latest advancements and studies in the field of Vitamin D research, particularly those published in the last 2-3 years. Including recent findings would make the review more current and relevant.
6. Lack of Critical Appraisal of Cited Studies: The paper often presents findings from various studies without critically appraising the quality of these studies. An assessment of the study designs, sample sizes, and potential biases would strengthen the review's conclusions.
In conclusion, while the paper provides a valuable overview of Vitamin D's role in fatigue mitigation, addressing these critiques would significantly enhance its comprehensiveness, accuracy, and relevance to current research and clinical practice.

English is fine
Author Response
After reviewing the paper titled "Vitamin D and its Role on the Fatigue Mitigation," several key areas for improvement have been identified. Below is a brief introduction followed by a list of critiques:
Introduction:
The paper delves into the multifaceted role of Vitamin D, primarily focusing on its impact in mitigating fatigue. While the authors have comprehensively reviewed various pathophysiological mechanisms and the involvement of Vitamin D, certain critical aspects seem to be overlooked or insufficiently explored.
- Lack of Research Methodology: The paper does not clearly outline the methodology used for the literature review. A detailed description of the databases searched, keywords used, inclusion and exclusion criteria, and the process of selecting articles would enhance the reliability and reproducibility of the review.
We thank the reviewer for noticing this aspect. To be clear and to not generate on the reader expectations about a systematic review, which is not our case, we modified the title and the text by specifying that this manuscript is a narrative review, thus not a systematic review (which otherwise would have required a specific section on methodologies, inclusion and exclusion criteria, that, on the contrary, are not requested for narrative reviews).
- Omission of Key Literature on Vitamin D and Musculoskeletal Pain: The paper fails to address significant studies linking Vitamin D deficiency with musculoskeletal pain, which is closely related to fatigue. Inclusion of such studies would provide a more holistic understanding of Vitamin D's role in mitigating fatigue-related symptoms.
We thank the reviewer for this comment. We added a paragraph on vitamin D and musculoskeletal pain in the revised version of the manuscript.
- Insufficient Analysis of Gender and Age Differences: While the paper touches upon different population groups, it lacks a detailed analysis of how Vitamin D’s role in fatigue mitigation varies across different genders and age groups. Considering the physiological differences in Vitamin D metabolism, such an analysis would be pertinent.
According to the reviewer’s request, we added information about gender specific differences where this information was available from the literature through the manuscript. While we agree with the reviewer that we did not discuss how fatigue mitigation varies across different genders in previous version of the paper, we would like to keep to the reviewer’s attention paragraph 3.5, that was specifically focused on fatigue in aging.
- Limited Discussion on Vitamin D Sources and Supplementation Guidelines: The paper does not sufficiently explore the different sources of Vitamin D (e.g., sunlight exposure, diet, supplements) and lacks specific guidelines or recommendations for Vitamin D supplementation, which would be valuable for both clinicians and patients.
We agree with reviewer 2 that this paper don’t address guidelines on vitamin D supplementation or sources. However, this is not the focus of the manuscript, which aims at describing potential implications of vitamin D in modulating diseases-associated fatigue. Moreover, in the literature there are a lot of dedicated manuscript aimed at thoroughly addressing these topics (eg. Vitamin D Sources, Metabolism, and Deficiency: Available Compounds and Guidelines for Its Treatment. doi: 10.3390/metabo11040255). For this reason, we introduced the main vitamin D sources in section 2 (lines 59-67) and then we underline that the unavailability of a specific guideline for supplementation as each study reported different doses to different physio-pathological conditions of patients (lines 324-332).
Inadequate Exploration of the Role of Vitamin D in Chronic Illnesses: While some chronic conditions are discussed, the paper does not thoroughly explore the role of Vitamin D in the context of chronic illnesses known for causing fatigue, such as chronic fatigue syndrome and fibromyalgia.
While we did not discuss all the chronic diseases that might benefit from vitamin D supplementation (which is out of the scope of this review), we added a dedicated paragraph on chronic fatigue syndrome as the reviewer’s suggested (section 3.7). Moreover, we further integrated the paragraph on fibromyalgia already present in the previous version of the manuscript (section 3.1).
- Neglect of Recent Advancements in Vitamin D Research: The paper does not incorporate the latest advancements and studies in the field of Vitamin D research, particularly those published in the last 2-3 years. Including recent findings would make the review more current and relevant.
We agree with the reviewer about the fact that we did not included all the manuscript recently published about research on vitamin D, which is a very broad topic if we considered it not only from the point of view of the fatigue. We reviewed the literature and added the following studies:
- Lombardo, M., Feraco, A., Ottaviani, M., Rizzo, G., Camajani, E., Caprio, M., & Armani, A. The efficacy of vitamin D supplementation in the treatment of fibromyalgia syndrome and chronic musculoskeletal pain. Nutrients, 2022. 14(15), 3010.
- Roy, N. M., Al-Harthi, L., Sampat, N., Al-Mujaini, R., Mahadevan, S., Al Adawi, S., ... & Qoronfleh, M. W. Impact of vitamin D on neurocognitive function in dementia, depression, schizophrenia and ADHD. FBL, 2020. 26(3), 566-611
- Krisanova, N., Pozdnyakova, N., Pastukhov, A., Dudarenko, M., Maksymchuk, O., Parkhomets, P., ... & Borisova, T. Vitamin D3 deficiency in puberty rats causes presynaptic malfunctioning through alterations in exocytotic release and uptake of glutamate/GABA and expression of EAAC-1/GAT-3 transporters. FCT, 2019. 123, 142-150.
- Goyal, V., & Agrawal, M. Effect of supplementation of vitamin D and calcium on patients suffering from chronic non-specific musculoskeletal pain: a pre-post study. Journal of Family Medicine and Primary Care, 2021. 10(5), 1839.
If the reviewer thinks that there are specific manuscripts that should be cited because they can contribute to the specific topic of fatigue modulation by vitamin D, we are happy to accept the suggestion and to add them in the manuscript.
- Lack of Critical Appraisal of Cited Studies: The paper often presents findings from various studies without critically appraising the quality of these studies. An assessment of the study designs, sample sizes, and potential biases would strengthen the review's conclusions.
In conclusion, while the paper provides a valuable overview of Vitamin D's role in fatigue mitigation, addressing these critiques would significantly enhance its comprehensiveness, accuracy, and relevance to current research and clinical practice.
To improve the comprehensiveness of the manuscript, we thoroughly reviewed the entire article by clarifying mechanisms of action of vitamin D and improved the adherence of the text with references reported. The revised version of the manuscript reports additional section on specific fatigue-related diseases and several newly included studies on the topics. We added details about the studies supporting the statements throughout the entire manuscript.
Reviewer 3 Report
Comments and Suggestions for Authors
The current manuscript is a good collection of the Vitamin D and its role on the fatigue mitigatio. However the authors did not use animated illustrations for easier understanding of the readers, need to add some tables with more concise information about the research done till now, findings, etc. Also there could be a future guideline for the researchers how and what to do in future to enhance the knowledge. The authors could also add a limitation of the research works done till now.
Comments on the Quality of English LanguageEnglish language seems to be good.
Author Response
The current manuscript is a good collection of the Vitamin D and its role on the fatigue mitigation.
However the authors did not use animated illustrations for easier understanding of the readers, need to add some tables with more concise information about the research done till now, findings, etc.
We thank the reviewer for this suggestion. We added two figures where we summarized the influence of vitamin D on several metabolic pathways and the main physio-pathological states on which we focused studying the impact of hypovitaminosis.
Also there could be a future guideline for the researchers how and what to do in future to enhance the knowledge. The authors could also add a limitation of the research works done till now.
We thank the reviewer for this suggestion. We modified the conclusion paragraph adding a disclosure on limitations of the current knowledge and studies on this topic and a recommendation to the scientific community to conduct RCT, which are essential to really document the efficacy of vitamin D supplementation in the context of fatigue mitigation.
Round 2
Reviewer 1 Report
Comments and Suggestions for Authors
The author gave the appropriate responses and corrections to my review comments.
Author Response
In attachment the answers to referee n. 2
Reviewer 2 Report
Comments and Suggestions for Authors
The revised manuscript presents a significant improvement over the previous version, offering a comprehensive and insightful exploration of Vitamin D's role in fatigue mitigation. The authors have effectively addressed many of the initial concerns. However, to further enhance the manuscript's quality and academic rigor, it is recommended that the authors align their review with the SANRA (Scale for the Assessment of Narrative Review Articles) guidelines. This will ensure a more systematic and transparent approach in the narrative review process.
Specific Comments as per SANRA:
Abstract and Keywords:
The revised abstract provides a clearer overview of the manuscript. Including a brief mention of the methodological approach would align it with SANRA guidelines.
Keywords are well-chosen and relevant.
Introduction:
The introduction is now more comprehensive, setting a clear context for the review. A brief mention of the research gap the review aims to address would further align with SANRA standards.
Methodology:
The methodology section has improved, but further details on the literature search strategy (including databases, keywords, and inclusion/exclusion criteria) are needed for compliance with SANRA guidelines.
Content:
The content is thorough and well-presented. Adding a section that critically evaluates the literature, discussing both strengths and limitations of the included studies, would be beneficial.
Discussion and Conclusion:
The discussion and conclusion sections are well-articulated. Emphasizing the implications of the findings and addressing potential research biases would enhance these sections as per SANRA recommendations.
Author Response

(The authors gave the same response as above.)

Reviewer 3 Report
Comments and Suggestions for Authors
The manuscript is improved. Thank you very much for the quick revision of the manuscript.
Comments on the Quality of English LanguagePlease check for the English language and improve as much as possible.
Author Response

(The authors gave the same response as above.)
